# Examining Anti-Poverty Programs to Address Student's Unmet Basic Needs at Texas Hispanic-Serving Institutions over the Course of the COVID-19 Pandemic

Lisa K. Zottarelli [1,*], Thankam Sunil [2], Xiaohe Xu [3] and Shamatanni Chowdhury [2]

[1] College of Social Work, University of Tennessee, Knoxville, TN 37996, USA
[2] Department of Public Health, University of Tennessee, Knoxville, TN 37996, USA; tsunil@utk.edu (T.S.); schowd15@vols.utk.edu (S.C.)
[3] College of Health, Community and Policy, University of Texas at San Antonio, San Antonio, TX 78249, USA; xiaohe.xu@utsa.edu
* Correspondence: lzottare@utk.edu

**Abstract:** Many post-secondary institutions have implemented anti-poverty programs to address students' basic needs insecurities. This study examined the provision of 17 types of basic needs programs at Texas Hispanic-serving institutions over the course of the COVID-19 pandemic with the aim to identify changes in the number and types of programs offered as well as factors that may influence the presence of specific types of basic needs programs on campus. While the average number of basic needs programs per institution varied little over time, the specific types of programs that were offered changed. Institution type as a 2-year or 4-year institution was associated with providing on-campus mental health services, on-campus physical health services, and after-school care for students' children at pre-pandemic and anticipated post-pandemic time points and employing students and free food or meal vouchers at the pre-pandemic time point. The percentage of students receiving Pell Grants was associated with basic needs programs to assist students applying for public services and referrals to off-campus health services pre-pandemic and anticipated post-pandemic. The presence of an on-campus free food pantry was associated with the percentage of students receiving Pell Grants at the anticipated post-pandemic time point only. Over the course of the pandemic, there were changes to the types of basic needs programs offered. Some types of basic needs programs were associated with institutional and/or student characteristics. Given the continued presence of basic needs programs through the course of the pandemic and into the post-pandemic period, the use of these kinds of programs and services to support students, while influenced by external factors such as the pandemic, appears institutionally established as a way to facilitate going to college for students in need.

**Keywords:** student support; student success; food insecurity; housing insecurity; emergency aid

## 1. Introduction

In the decade prior to the COVID-19 pandemic, there had been growing awareness of the prevalence of basic needs insecurities among students at colleges and universities across the United States. Amid calls to address the economic plight faced by many students within U.S. post-secondary institutions and the results of data gathered by colleges and universities about their students' food and housing insecurities [1], many post-secondary institutions developed new or reconceptualized existing anti-poverty programs to increase the retention and completion among economically disadvantaged students. Often referred to as basic needs initiatives (BNI), these anti-poverty programs sought to reduce non-academic barriers to college persistence and completion [2]. The array of services under the umbrella of BNI included food pantries, emergency housing, emergency financial assistance, childcare, transportation, clothing, and healthcare [1,3]. The COVID-19 pandemic prompted abrupt

changes to basic needs programs [4]. While the trends in awareness and response to student needs started before the pandemic, changes made during the pandemic, as well as post-pandemic economic conditions, have magnified wellness and basic needs insecurities.

The COVID-19 pandemic was a significant disruption to higher education with one of the defining concepts being pivot. Pivot encapsulated the rapid changes and adjustments made in response to new information and new conditions. The pivot took many forms but most significantly referenced changes from conventional in-person instruction and service delivery to remote (i.e., off-campus) or some variation of remote teaching and services. As the pandemic continued, pivots between remote and in-person and variations on in-person instruction and services continued. The pandemic and the necessitated pivots required that institutional priorities be allocated to preserve essential functions and academic continuity [5,6]. For many post-secondary institutions, the challenges of the pandemic were compounded by declining enrollment, budget shortfalls, and increased expenses [7].

While the pandemic created challenges for all post-secondary institutions, some institutions, including Hispanic-serving institutions (HSIs), were particularly disadvantaged during the pandemic. HSIs are often less well-resourced than other colleges and universities [8], and the policies enacted during the COVID-19 pandemic prioritized criteria that often further disadvantaged HSIs [9]. Scientific and technical programs, which include programs offered at many HSI 2-year institutions that require hands-on practice and lab components, faced significant challenges in the pivot to remote instruction and service delivery [10]. Regardless of program or course content, the switch to remote learning environments placed additional technological pressures on students. Greater proportions of students at HSIs reported unstable internet quality and access and a lack of device reliability and adequacy compared to students at other types of post-secondary institutions [11].

Addressing student basic needs aligns with institutional student success and completion priorities [12–14] and has been part of a broader trend in higher education to address student well-being and basic needs insecurity [4,15]. The pandemic pivots and subsequent economic and enrollment conditions were disruptive for many broad-access HSIs, especially community colleges [8]. In the post-pandemic, there is growing attention to the costs of attending college and the financial challenges faced by many students as they pursue post-secondary education. As institutions respond to the basic needs insecurities in a post-pandemic environment, it is important to understand how the pandemic shifted services and programs directed at student's unmet basic needs. It is unclear if basic needs programs offered prior to the pandemic were determined to be essential services during the pandemic and, therefore, necessary to operational and academic continuity. It is critical to have a clear understanding of the changes that occurred to types of basic needs programs provided by institutions over the course of the pandemic because this provides insight into post-pandemic anti-poverty programs that are designed to support student retention and completion. Further, if these basic needs programs were anti-poverty in intention, it would be expected that student economic needs would indicate the presence of specific basic needs programs at post-secondary institutions. Our research considers the following questions: What changes occurred in BNI offerings over the course of the pandemic at Texas Hispanic-serving institutions? Were specific types of basic needs programs associated with institutional and/or student characteristics?

## 2. Context of Texas Hispanic-Serving Institutions

Our study was conducted on Texas HSIs. The data collected were part of a funded research project that sought to provide program evaluation training to examine relationships between basic needs programs and student success in terms of retention, persistence, and graduation. Texas is a large and diverse state with a complex system of public post-secondary institutions. It has numerous institutions that meet the Department of Education's HSI designation by enrolling at least 25% Latinx full-time equivalent undergraduates. Texas postsecondary institutions are governed by a single higher education coordinating

board, have to meet uniform state legal requirements and funding models and share the same institutional accreditor.

We chose to focus our project on HSIs because the senior researchers had spent much of their careers at HSIs, and we are committed to supporting the missions of many HSIs to support their students and communities. Additionally, as the number of institutions with the designation of HSI increases, understanding differences in institutional environments and organizational behavior within this broad category of institutions is important [16]. In Texas, HSIs differ markedly and include community colleges, four-year primarily undergraduate institutions, and research-intensive institutions with numerous graduate programs. HSIs range from small institutions with student enrollment of fewer than 5000 to large institutions with enrollments of more than 20,000 students. Some HSIs in Texas are residential, while others are commuter campuses. Many of these institutions have significant roles in the community and are viewed as critical to building local human and economic capital [16]. In Texas in 2021–2022, 42.5% of the undergraduate headcount was Hispanic, and many of these students were enrolled in HSIs [17].

Enrollment at Texas public institutions has declined over the course of the pandemic. In the 2018–2019 academic year, there were 1,570,539 students enrolled in Texas nonprofit higher education institutions [18]. In the first full academic year of the pandemic, 2020–2021, enrollment declined to 1,509,761 [19] and then continued to decline in 2021–2022 to 1,490,079 students [20]. Before the pandemic, the Texas Higher Education Coordinating Board [18–20] found that students attending Texas post-secondary institutions rely extensively on federal financial assistance and experience significant unmet economic needs every year. Over this time period, approximately 20% of unduplicated financial aid recipients were from households with income below the poverty line. Over a third of unduplicated financial aid recipients were from households under the median income. The state had the second highest food insecurity rate in the United States, which had increased in 2022 to almost 1 in 6 households being food insecure [21]. There was evidence of unmet student basic needs on campus prior to and during the pandemic [22–26]. Given the economic conditions of many students' households throughout the pandemic and into the post-pandemic, it is reasonable to expect that unmet needs will continue to need to be addressed.

### 3. Basic Needs Initiatives as Anti-Poverty Programs to Meet Unmet Student Needs

There are significant benefits to earning a college degree, including greater lifetime earnings [27,28]. Yet, with deepening state cuts in funding for higher education and increasing tuition costs, it is more difficult for students, especially students from lower-income backgrounds, to enroll, persist, and graduate [29]. For students who are classified as independent (i.e., not dependent upon parents based on the federal financial aid definition regardless of the actual support received), 74% of students without dependents and 80% of students with dependents had incomes 200% below the poverty guidelines [30]. Many students struggle to meet their basic needs while in college, and an inability to meet their basic needs can force academically able students to stop out or drop out.

Basic needs programs and the students they support face many challenges, including stigma [31] and a lack of programs and funding sufficient to address the unmet needs [2,4,31]. Based on research at post-secondary institutions across the United States, there is a high prevalence of economic and health insecurities among college students. The most frequently studied basic needs insecurities are food and housing insecurity, and the prevalence of food and housing insecurities is high across the country. Of the 86,000 college and university students across 123 institutions surveyed in the fall of 2018, 45% had experienced food insecurity in the previous 30 days, and 56% had experienced housing insecurity in the previous year [32]. In Texas, students at many institutions reported food insecurity prior to the pandemic [22,23], and the prevalence of food insecurity remained high during the pandemic [26,33].

Unmet needs, in conjunction with economic and health insecurities, make attending college more difficult and increase the likelihood of leaving higher education for non-academic reasons. That is, a lack of resources to meet basic needs places students at greater risk of experiencing difficulties attending classes and performing to their full academic ability [34–37]. For example, students who experience food insecurity have greater odds of being in the lowest 10% of grade point averages (GPAs), lower odds of being in the upper 10% of GPAs [38], and have lower overall GPAs [36,39]. Furthermore, food insecurity may be related to poor mental health outcomes among college students [34,36], which also hinders students' academic performance. A high proportion of college students have unmet mental health needs that increase the odds of stopping out, and few students receive even minimally adequate treatment [40]. To summarize, experiencing unmet basic needs while in college impairs students' ability to succeed in school, which in turn diminishes the likelihood of earning a degree.

Pell Grants, intended for economically disadvantaged students and based on expected family contribution, on average, cover less than one-third of the costs of attendance for students not living with family [41]. This places significant financial strain on covering college costs and basic needs. Under these financial conditions, students may stop out or drop out of college. Yet, when financial support adequately covers tuition, housing, and food, thereby addressing the need for money, students from economically disadvantaged backgrounds succeed in college [42]. BNIs are a means by which institutions attempt to respond to this gap in students' financial resources by addressing unmet basic student needs.

To address the prevalence of economic insecurity resulting in unmet basic needs among students, many colleges and universities launched programs designed to address poverty on campus. These have taken the form of improving access to food, housing, childcare, transportation, etc. Many colleges and universities operate food pantries [3,43,44] and numerous other programs and services designed to address basic needs, including emergency financial support, transportation, childcare, mental and physical health services, and clothing [3]. Students use campus food pantries to improve food security [26].

There was a growing practice of implementing basic needs programs to address a variety of unmet student needs prior to the COVID-19 pandemic, with most programs directed at providing services in person and on campus. The COVID-19 pandemic disrupted higher education in fundamental ways, including the delivery of instruction and how students engage with their institutions. It also exposed the precarity of many students. The COVID-19 pandemic and subsequent economic downturns have worsened economic situations and increased the unmet needs of many college students [33,43]. There were shifts in community college basic needs programs that included increases in emergency financial support and decreasing services that required direct contact, such as childcare programs and clothing closets [3].

It is relevant to examine basic needs initiatives to address unmet student basic needs at a state level and align them to institutional characteristics such as HSI status. The state in which public institutions are located influences institutional policy, priorities, and funding. For example, California has examined both student needs and institutional responses to those needs across university systems [2,12]. California has policies, budget allocations, and coordinated support of basic needs programs on college campuses across the state [45,46]. The State University of New York and The City University of New York are engaging in system-wide efforts to address a range of basic needs programs [47]. New Jersey has launched a state-wide basic needs resource page for college students [48]. Texas, in contrast, lacks this kind of organized approach, but institutions are encouraged to engage in efforts to support student success. One approach taken by many Texas colleges and universities has been to develop programs directed at supporting specific student populations, including lower-income and first-generation college students. In the absence of state-level initiatives, institutions in states like Texas are still implementing basic needs programming. Understanding the types and prevalence of anti-poverty basic

needs programs where they are institutionally driven may provide insights that differ from state-driven initiatives.

In Texas, most community colleges and broad-access regional public institutions are HSIs. Many of the students attending HSIs are from the county or neighboring counties where the institution is located. As such, the HSIs have a significant role in the community and have a student body reflective of that community [16]. Further, HSIs are more likely to enroll first-generation, lower-income, historically underrepresented, and non/post-traditional students [49,50]. These students are more likely to experience basic needs insecurities while in college.

Given Texas' lack of a state-wide or postsecondary system-wide basic needs initiatives for college students, the economic situation of many Texas post-secondary students, and the importance of HSIs in providing post-secondary educational opportunities to local students, we first examined the prevalence of basic needs programs at Texas HSIs as we answered the question: *What changes occurred in BNI offerings over the course of the pandemic at Texas Hispanic-serving institutions?* To do so, we explored changes in the number and types of basic needs programs offered before and during the pandemic. Then, we examined changes to the number and types of basic needs programs that institutions planned to support after the pandemic.

Next, we examined two factors that may influence the presence of specific basic needs programs at post-secondary institutions. We focused on one institutional factor: the classification of the institution as a 2-year or 4-year institution. This is relevant given the differences in student body characteristics and the funding sources between the two institution types. We used the percentage of students receiving Pell Grants as the student-focused factor, given that it indicates the economic conditions of the student body. We asked the question: *Were specific types of basic needs programs associated with institutional and/or student characteristics?* This question was answered through the testing of two hypotheses for each type of basic needs program identified in the study.

**Hypotheses 1:** *There will be a relationship between institution type (e.g., 2-year and 4-year and the presence of [specific basic needs program].*

**Hypothesis 2:** *There will be a relationship between the percentage of students receiving Pell grants and the presence of [specific basic needs program].*

## 4. Methods and Materials

We report on a study of basic needs programs conducted during the pandemic that focused on identifying the array of basic needs programs offered at post-secondary HSIs in Texas. Data for this study come from an online survey of college and university administrators at Texas HSIs that was conducted to gather information about the types of basic needs programs offered at their institutions prior to and during the pandemic and the anticipated programs to be offered after the pandemic. Administrators were selected from Texas post-secondary institutions designated HSI or listed on the Hispanic Association of Colleges and Universities (HACU) list of HSIs. In total, 96 Texas colleges and universities met these criteria.

## 5. Participants and Procedures

Four administrators at each Texas HSI were identified from a search of the institution's website. We collected the names and work email addresses of vice presidents, deans, and directors of student services and equivalent positions. The first person from each institution was invited to participate in the study via an email sent from the lead researcher. The initial email described the study and included a link to the survey hosted on the lead author's institutional Qualtrics site. If no response was received, two follow-up emails were sent approximately one week apart. If no response was received after the third attempt at contact, a different administrator at the institution was contacted via email, inviting them

to participate and follow the same procedure. In some cases, an email would bounce back indicating the email address was not valid, or the person contacted would respond to the email invitation directing that another person at the institution be contacted. Up to four administrators at each institution were contacted. Only one response was collected per institution. Of the 96 institutions contacted, 46 (47.9%) had an institutional administrator complete the survey. Data were collected from 22 October to 8 December 2020, the first fall semester of the COVID-19 pandemic.

## 6. Survey Instrument and Institutional Data

The survey consisted of a series of close- and open-ended questions. The instrument was divided into two parts. The first part focused on specific basic needs initiatives and included nine thematic areas: food, transportation, housing, financial assistance, clothing, childcare, mental and physical health services, social service referrals, and employment. The themes identified were found in the academic and practice literature on basic needs programs. Respondents were asked about a total of 17 basic needs programs within the nine themes. For each basic needs item, respondents were asked a set of three questions. First, "Prior to the pandemic, did your institution have [basic needs program] for students in need?" Response categories were "yes", "no", and "not sure". This was followed by the question, "This semester, do you currently have [basic needs program] for students in need?" Response categories were "yes", "no", and "not sure". Then, participants were asked, "In the future, do you anticipate providing [basic needs program] to students in need?" The response categories were "definitely yes", "probably yes", "probably not", and "definitely not".

In the second part of the survey, participants were also asked to provide their institutional affiliation. This information allowed the survey data to be matched with institutional data. The percentage of students receiving Pell Grants was used to gauge the overall economic status and financial needs of the enrolled students. Data came from the institutional factbooks provided on the institutional websites. Data on classification as 2-year and 4-year institutions and additional information, such as total enrollment and graduation rates, were from the Texas Higher Education Coordinating Board [19]. The Institutional Review Boards (IRB) at the University of Tennessee, Knoxville (UTK) and the University of Texas at San Antonio determined that the study did not require IRB oversight and review as defined in 45 CFR 46.102 (3)(1) (UTK IRB-20-06081-XP).

### 6.1. Sample of Texas Hispanic-Serving Institutions

Forty-three of the 46 institutions with administrators who responded to the survey provided their institution's name. The 43 institutions where it was possible to link survey responses with institutional data were compared to the population of Texas HSIs. The results showed no statistically significant differences in total enrollment ($t = 1.58$ (94), $p = 0.12$), graduation rates ($t = 1.25$ (94), $p = 0.22$), and percentage of students receiving Pell Grants ($t = -1.01$ (94), $p = 0.32$). Of the institutions where an administrator completed the survey, 18 (41.9%) were 2-year institutions, and 25 (58.1%) were 4-year institutions.

### 6.2. Analyses

The response categories for the pre-pandemic and during-pandemic time points were recoded to 1 for "yes" and 0 for "no" or "not sure". The response categories for the anticipated offering of the basic needs programs post-pandemic were recoded as 1 for "definitely yes" and 0 for the remaining responses. For the total number of basic needs programs at each time point, the responses were added ranging from 0 to 17. The mean, standard deviation, and percent change are reported for the total numbers at each time point. Frequencies, percentages, and relative percent changes were calculated for the specific basic needs program categories across the three time points.

The institutions were divided evenly into three categories based on the percentage of their students receiving Pell. Specifically, institutions with less than 27% of their students

receiving Pell (*n* = 14, 33.3%), institutions with between 28% and 39% of students on Pell (*n* = 14, 33.3%), and institutions with 40% or more of students receiving Pell (*n* = 14, 33.3%). Chi-square was used to test the association between the percentage of students receiving Pell Grants and specific basic needs programs offered and between institution type, 2-year or 4-year, and the specific basic needs programs offered. IBM SPSS version 27 was used for statistical analysis [51].

## 7. Results

### 7.1. Changes in Basic Needs Services between Pre-Pandemic and Pandemic

First, we examined changes in the number of basic needs programs per institution prior to and during the pandemic. Before the pandemic, the institutions offered a mean of 10.4 (*SD* = 2.9) basic needs programs. Employ students, community-based social service agency referrals, and on-campus mental health services were the most frequently offered, with more than 90% of the institutions in the sample offering programs in these areas. In addition to these programs, three additional areas (i.e., mental health referrals to low-cost off-campus services, physical health referrals to low-cost off-campus services, and on-campus free food pantry) were offered at more than 75% of the institutions. Fewer than a quarter of the institutions offered subsidized meal plans, after-school care for students' children, and free "everyday" clothing.

Table 1 shows the first pivot from pre-pandemic basic needs programs to programs offered during the pandemic. During the first fall semester of the pandemic, the institutions offered a mean of 10.2 (*SD* = 2.7) basic needs programs. Four types of basic needs programs (i.e., on-campus mental health services, physical health referrals to low-cost off-campus services, help applying for public services such as WIC or SNAP, and subsidized meal plans) were offered at the same number of institutions at the two time points. Three types of programs were offered at more than 90% of the institutions (i.e., community-based social service referrals, employed students, and on-campus mental health services) prior to the pandemic. During the pandemic, mental health referrals to low-cost off-campus services also increased to being offered at more than 90% of the institutions.

**Table 1.** Number of Institutions Offering Specific Basic Needs Programs Before and During the Pandemic.

| Basic Needs Programs | Before Pandemic Number | Before Pandemic Percent | During Pandemic Number | During Pandemic Percent | Relative Percent Change |
|---|---|---|---|---|---|
| **Number and Percent of Institutions Offering Basic Needs Program** | | | | | |
| *Percent Change Increase in Number of Institutions Offering BNI* | | | | | |
| Emergency Financial Assistance | 32 | 74.4% | 37 | 86.0% | +15.6% |
| Free Meals or Food Vouchers | 19 | 44.2% | 21 | 48.8% | +10.5% |
| Mental Health Referrals to Low-cost Off-Campus Services | 38 | 88.4% | 40 | 93.0% | +5.3% |
| On-campus Physical Health Services | 22 | 51.2% | 23 | 53.5% | +4.5% |
| Community-based Social Services Agency Referrals | 41 | 95.3% | 42 | 97.7% | +2.4% |
| Employ Students | 41 | 95.3% | 42 | 97.7% | +2.4% |
| *No Change in Number of Institutions Offering BNI* | | | | | |
| On-campus Mental Health Services | 39 | 90.7% | 39 | 90.7% | 0% |
| Physical Health Referrals to Low-cost Off-campus Services | 36 | 83.7% | 36 | 83.7% | 0% |
| Help Applying for Public Services such as WIC or SNAP | 22 | 51.2% | 22 | 51.2% | 0% |
| Subsidized Meal Plans | 5 | 11.6% | 5 | 11.6% | 0% |
| *Percent Change Decrease in Number of Institutions Offering BNI* | | | | | |
| Preschool for Students' Children | 20 | 46.5% | 15 | 34.9% | −25.0% |
| Free Business Clothing for Job Interviews or Jobs | 29 | 67.4% | 22 | 51.2% | −24.1% |
| After-school Care for Students' Children | 7 | 16.3% | 6 | 14.0% | −14.3% |
| Free "Everyday" Clothing | 8 | 18.6% | 7 | 16.3% | −12.5% |
| On-campus Free Food Pantry | 36 | 83.7% | 33 | 76.7% | −8.3% |
| Free or Subsidized Transportation to and From Campus | 29 | 67.4% | 27 | 62.8% | −6.9% |
| Emergency Housing Assistance | 24 | 55.8% | 23 | 53.5% | −4.2% |

Six basic needs program types had an increase in the number of institutions offering the programs between the two time points. Emergency financial assistance increased by 15.6%, with a total of 37 (86.0%) offering this type of basic needs program during the pandemic. Free meals or food vouchers increased by 10.5%. Smaller increases were found in two other types of programs. There were decreases in the number of institutions offering seven types of basic needs programs. Two areas of decreased programming at institutions were in childcare for students' children and clothing. There was a 14.3% reduction in the after-school care for students' children and a 25.0% reduction in preschool programs during the pandemic. The offering of business clothing programs decreased by 24.1%, and free "everyday" clothing programs decreased by 12.5%. Additionally, the number of institutions offering an on-campus free food pantry decreased by 8.3%.

### 7.2. Anticipated Changes to the Post-Pandemic Basic Needs Services

Next, we considered the plans that college and university administrators shared about post-pandemic basic needs programs on their campuses. Table 2 shows the number of institutions that planned to offer specific basic needs programs and how the number of institutions planning to provide the specific programs changed from prior to the pandemic. Four types of basic needs programs (i.e., employ students, community-based social service agency referrals, on-campus mental health services, and on-campus physical health services) showed no change in the number of institutions from prior to post-pandemic. There were four types of basic needs programs (i.e., free "everyday" clothing, free meals or food vouchers, help applying for public services such as WIC or SNAP, and on-campus free food pantry) where more institutions planned to offer the BNI compared to before the pandemic. Nine types of basic needs programs were expected to decrease in the number of institutions, and these included basic needs programs that had increased during the pandemic. Emergency financial assistance had been offered at 32 (74.4%) of institutions prior to the pandemic, and this increased to 37 (86.0%) institutions during the pandemic but was expected to be offered at 30 (69.8%) institutions post-pandemic for a net decrease of 6.3%. Referrals to off-campus low-cost mental health providers follow a similar pattern, with an increase in institutions from before the pandemic (n = 38, 88.4%) to 40 (93.0%) institutions during the pandemic, followed by an anticipated decrease after the pandemic (n = 37, 86%). Three types of basic needs programs, emergency housing assistance, free or subsidized transportation, and after-school childcare for students' children, showed a decline in the number of institutions providing these services across the three points.

### 7.3. Associations between Institution Type and Specific Basic Needs Programs

Given the changes in the number of institutions offering specific basic needs programs, we examined institution type and student need to identify factors that might lead to decisions to offer specific basic needs programs (see Table 3). Chi-square was run with institution type (i.e., 2-year and 4-year institutions) and the specific basic needs programs offered before the pandemic and intended to be offered after the pandemic. Prior to the pandemic, there were statistically significant associations identified between five specific basic needs programs and institution types. These were free meals or vouchers for food, on-campus mental health services, on-campus physical health services, after-school care for students' children, and employment for students. Post-pandemic, on-campus physical health services and on-campus mental health services were anticipated to be offered at the same number of institutions as prior to the pandemic. Additionally, the relationship between after-school care for students' children and institution type was statistically significant.

**Table 2.** Institutions Planning to Offer Specific Basic Needs Programs After the Pandemic.

| Basic Needs Programs | Planned to Offer | | Relative Percent Change from Before the Pandemic |
|---|---|---|---|
| | Number | Percent | |
| Increase in Number of Institutions Offering BNI from Before the Pandemic | | | |
| Free "Everyday" Clothing | 10 | 23.8% | +25.0% |
| Free Meals or Food Vouchers | 20 | 46.5% | +5.3% |
| Help Applying for Public Services such as WIC or SNAP | 23 | 53.5% | +4.5% |
| On-campus Free Food Pantry | 37 | 86.0% | +2.8% |
| No Change in Number of Institutions Offering BNI | | | |
| Employ Students | 41 | 95.3% | 0 |
| Community-Based Social Services Agency Referrals | 41 | 95.3% | 0 |
| On-campus Mental Health Services | 39 | 90.7% | 0 |
| On-campus Physical Health Services | 22 | 51.2% | 0 |
| Decrease in Number of Institutions Offering BNI from Before the Pandemic | | | |
| Mental Health Referrals to Low-cost Off-campus Low Services | 37 | 86.0% | −2.6% |
| Physical Health Referrals to Low-cost Off-campus Low Services | 35 | 81.4% | −2.8% |
| Emergency Financial Assistance | 30 | 69.8% | −6.3% |
| Emergency Housing Assistance | 20 | 46.5% | −16.7% |
| Free Business Clothing for Job Interviews or Jobs | 24 | 55.8% | −17.2% |
| Subsidized Meal Plans | 4 | 9.3% | −20.0% |
| Preschool Care for Students' Children | 16 | 37.2% | −20.0% |
| Free or Subsidized Transportation to and From Campus | 22 | 51.2% | −31.8% |
| After-school Care for Students' Children | 5 | 11.6% | −40.0% |

**Table 3.** Association between Institutional Type and Specific Basic Needs Programs.

| Basic Needs Program Category | | 2 Year Institution | | 4 Year Institution | | Chi-Square (df = 1) |
|---|---|---|---|---|---|---|
| | | Count | Percent | Count | Percent | |
| On-campus Free Food Pantry | Prior | 14 | 77.8% | 22 | 88.0% | 0.802 |
| | After | 15 | 83.3% | 22 | 88.0% | 0.190 |
| Free Meals or Vouchers | Prior | 11 | 61.1% | 8 | 33.3% | 3.204 * |
| | After | 11 | 61.1% | 9 | 36.0% | 2.652 |
| Subsidized Meal Plans | Prior | 3 | 16.7% | 2 | 8.0% | 0.765 |
| | After | 3 | 16.7% | 1 | 4% | 1.990 |
| Transportation | Prior | 11 | 61.1% | 18 | 72.0% | 0.565 |
| | After | 11 | 61.1% | 11 | 44% | 1.226 |
| Emergency Housing | Prior | 11 | 61.1% | 13 | 52% | 0.352 |
| | After | 10 | 55.6% | 10 | 40% | 1.018 |
| Help Applying for Public Services such as WIC or SNAP | Prior | 11 | 61.1% | 11 | 45.8% | 0.963 |
| | After | 12 | 66.7% | 11 | 44% | 2.161 |
| Refer to Community Services | Prior | 17 | 94.4% | 24 | 96% | 0.057 |
| | After | 17 | 94.4% | 24 | 96% | 0.057 |
| Emergency Financial Assistance | Prior | 12 | 66.7% | 20 | 80% | 0.977 |
| | After | 12 | 66.7% | 18 | 72% | 0.141 |
| Free "Everyday" Clothing | Prior | 3 | 16.7% | 5 | 20% | 0.077 |
| | After | 5 | 27.8% | 5 | 20% | 0.355 |
| Business Clothing | Prior | 11 | 61.1% | 18 | 72% | 0.565 |
| | After | 10 | 55.6% | 14 | 56% | 0.001 |
| On-campus Mental Health Services | Prior | 18 | 100% | 21 | 84% | 3.175 * |
| | After | 18 | 100% | 21 | 84% | 3.175 * |
| Mental Health Referrals to Low-cost Off-campus Services | Prior | 16 | 88.9% | 22 | 95.7% | 0.681 |
| | After | 15 | 83.3% | 22 | 88% | 0.190 |
| On-campus Physical Health Services | Prior | 6 | 33.3% | 16 | 64.0% | 3.939 ** |
| | After | 6 | 33.3% | 16 | 64.0% | 3.939 ** |
| Physical Health Referrals to Low-cost Off-campus Services | Prior | 14 | 77.8% | 22 | 88% | 0.802 |
| | After | 14 | 77.8% | 21 | 84% | 0.268 |

**Table 3.** *Cont.*

| Basic Needs Program Category | | 2 Year Institution | | 4 Year Institution | | Chi-Square (df = 1) |
|---|---|---|---|---|---|---|
| | | Count | Percent | Count | Percent | |
| Preschool Childcare | Prior | 10 | 55.6% | 10 | 40% | 1.018 |
| | After | 8 | 44.4% | 8 | 32% | 0.694 |
| After-school Care for Students' Children | Prior | 6 | 33.3% | 1 | 4.0% | 6.607 ** |
| | After | 4 | 22.2% | 1 | 4.0% | 3.382 * |
| Employ Students | Prior | 16 | 88.9% | 25 | 100% | 2.913 * |
| | After | 17 | 94.4% | 24 | 96% | 0.057 |

* $p < 0.10$, ** $p < 0.05$.

### 7.4. Association between Percentage of Students on Pell and Specific Basic Needs Programs

During pre- and post-pandemic periods, three types of basic needs programs had statistically significant associations with the percentage of students receiving Pell Grants at an institution (see Table 4). These were on-campus free food pantries, help applying for public services such as WIC or SNAP, and physical health referrals to low-cost off-campus services. Only help applying for public services was significantly associated with the percentage of students receiving Pell Grants post-pandemic.

**Table 4.** Association between Percentage of Students Receiving Pell Grants and Specific Basic Needs Programs.

| Basic Needs Program Type | | Percentage of Students Receiving Pell Grant | | | | | | Chi-Square (df = 2) |
|---|---|---|---|---|---|---|---|---|
| | | 27% or Fewer | | 28–39% | | 40% or More | | |
| | | Count | % | Count | % | Count | % | |
| On-campus Free Food Pantry | Prior | 9 | 64.3% | 13 | 92.9% | 13 | 92.9% | 5.486 |
| | After | 9 | 64.3% | 14 | 100% | 13 | 92.9% | 8.167 * |
| Free Meals or Vouchers | Prior | 5 | 38.5% | 7 | 50% | 7 | 50% | 0.475 |
| | After | 5 | 35.7% | 8 | 57.1% | 7 | 50% | 1.336 |
| Subsidized Meal Plans | Prior | 2 | 14.3% | 2 | 14.3% | 1 | 7.1% | 0.454 |
| | After | 1 | 7.1% | 2 | 14.3% | 1 | 7.1% | 0.553 |
| Transportation | Prior | 9 | 64.3% | 8 | 57.1% | 11 | 78.6% | 1.500 |
| | After | 8 | 57.1% | 6 | 42.9% | 8 | 57.1% | 0.764 |
| Emergency Housing | Prior | 8 | 57.1% | 7 | 50% | 8 | 57.1% | 0.192 |
| | After | 7 | 50% | 6 | 42.9% | 6 | 42.9% | 0.192 |
| Help Applying for Public Services | Prior | 4 | 28.6% | 5 | 35.7% | 12 | 92.3% | 13.005 * |
| | After | 7 | 50% | 4 | 28.6% | 12 | 85.7% | 9.419 * |
| Community-based Social Service Agency Referrals | Prior | 12 | 85.7% | 14 | 100% | 14 | 100% | 4.200 |
| | After | 12 | 85.7% | 14 | 100% | 14 | 100% | 4.200 |
| Emergency Financial Assistance | Prior | 9 | 64.3% | 11 | 78.6% | 11 | 78.6% | 0.985 |
| | After | 10 | 71.4% | 10 | 71.4% | 9 | 64.3% | 0.223 |
| Free "Everyday" Clothing | Prior | 4 | 28.6% | 2 | 14.3% | 2 | 14.3% | 1.235 |
| | After | 4 | 28.6% | 3 | 21.4% | 3 | 21.4% | 0.263 |
| Business Clothing | Prior | 8 | 57.1% | 11 | 78.6% | 9 | 64.3% | 1.500 |
| | After | 7 | 50% | 10 | 71.4% | 3 | 42.9% | 2.499 |
| On-campus Mental Health Services | Prior | 13 | 92.9% | 13 | 92.9% | 12 | 85.7% | 0.553 |
| | After | 13 | 92.9% | 13 | 92.9% | 12 | 85.7% | 0.553 |
| Mental Health Referral to Low-cost Off-campus Services | Prior | 11 | 91.7% | 13 | 92.9% | 13 | 92.9% | 0.017 |
| | After | 11 | 78.6% | 12 | 85.7% | 13 | 92.9% | 1.167 |
| On-campus Physical Health Services | Prior | 5 | 35.7% | 8 | 57.1% | 8 | 57.1% | 1.714 |
| | After | 4 | 28.6% | 8 | 57.1% | 9 | 64.3% | 4.000 |
| Physical Health Referrals to Low-cost Off-campus Services | Prior | 8 | 57.1% | 13 | 92.9% | 14 | 100% | 10.629 * |
| | After | 8 | 57.1% | 12 | 85.7% | 14 | 100% | 8.647 * |
| Preschool childcare | Prior | 6 | 42.9% | 7 | 50% | 6 | 42.9% | 0.192 |
| | After | 4 | 28.6% | 7 | 50% | 4 | 28.6% | 1.867 |
| After-school Care for Students' Children | Prior | 2 | 14.3% | 2 | 14.3% | 2 | 14.3% | 0.000 |
| | After | 1 | 7.1% | 2 | 14.3% | 1 | 7.1% | 0.553 |
| Employ students | Prior | 13 | 92.9% | 14 | 100% | 13 | 92.9% | 1.050 |
| | After | 13 | 92.9% | 14 | 100% | 13 | 92.9% | 1.050 |

* $p < 0.05$.

## 8. Discussion

Many of the student well-being and basic needs insecurities that were magnified during the pandemic were present prior to the pandemic [4,15]. More than a decade of research on basic needs insecurities, especially food and housing insecurities, as well as student experiences at colleges and universities, has led to the development of basic needs programs at specific institutions as well as state-wide initiatives. The post-pandemic economic conditions have continued to stress both students and postsecondary institutions and have continued to demonstrate the need for basic needs programs and services. Given the student populations, decreased state support, increased costs, and declining enrollment, postsecondary institutions must prioritize where resources are allocated. Basic needs programs, for the most part, were maintained and, in some cases, expanded during and after the pandemic, suggesting their importance to colleges and universities and their students. This supports previous research that has focused on the importance of basic needs programs as aligning with institutional success and completion priorities [12–14], even in states without state and system-wide initiatives.

This study adds to the literature on institutional responses to the basic needs insecurities of college students by exploring the presence of programs across institutions and through the COVID-19 pandemic. The pandemic and post-pandemic economic conditions increased the unmet needs of many college students [33,41]. This study provides information on the variety of programs and services that colleges and universities have developed to address non-academic and non-social aspects of students' lives at the most basic level of food, shelter, clothing, transportation, healthcare, and childcare. It also indicates formal programs to provide referrals to services within the community, which can be important when students may not be familiar with the community or services in the city where the college is located, not be aware of services, and/or not realize they might be eligible for services. It contextualizes these programs as anti-poverty programs designed to directly address poverty in the immediate term while a student is in college and in the long term through student retention and graduation. Some basic needs programs are more likely to be provided at institutions that demonstrate higher economic needs among their students and differ by institution type. This area of research is important as an examination of the institutional response to student basic needs and provides insight into the number and locations of such programs within the context of Hispanic-serving institutions, as well as institutional and student factors that are associated with the provision of basic needs programs and services.

### 8.1. What Changes Occurred in Basic Needs Offerings over the Course of the Pandemic?

Prior to the pandemic, Texas HSIs were providing an array of basic needs services for their students. Changes during the pandemic with the provision of certain types of basic needs programs appear to reflect pandemic conditions. The marked increase in the number of institutions offering emergency financial assistance may be linked to funding received through the Higher Education Emergency Relief Funds (HEERF) [52]. HEERF allocated over six billion dollars for emergency financial aid and relied on higher education institutions to distribute the funds to students [53]. Minority-serving institutions, including HSIs, were more likely to disperse the funds to students automatically without the requirement that students provide evidence of hardship [53]. HEERF funding ended, and this could explain the anticipated reduction in institutions offering emergency financial assistance post-pandemic.

Increases in referrals for mental and physical healthcare services in the community, along with referrals to community-based agencies, may have been a result of students being off campus due to the pivot to remote learning. Research conducted during the pandemic indicates that students experienced high levels of stress, anxiety, and depression during the pandemic [54–56]. The COVID testing and other physical health aspects of a pandemic may account for an increase in physical health services, both on-campus and through referral.

The decrease in the number of services may have been a result of both the need to pivot away from in-person services and services where contact was unavoidable, such as clothing closets. For the most part, where the decreases occurred were in the areas of programs where close contact or physical presence was required for the services to be provided, such as childcare and after-school care. This follows the broader trend of a lack of affordable quality childcare during the pandemic [57]. The reduction in these services would have affected student-parent populations but had less impact on the larger college student populations. Two of the basic needs programs often examined in the literature, food pantries and emergency housing, both saw a reduction. The shift to emergency financial aid and away from food and emergency housing aligns with the reduced presence of students on campus. Additionally, emergency housing was anticipated to continue to decline post-pandemic, leaving this basic need further unaddressed.

*8.2. Were Specific Types of Basic Needs Programs Associated with Institutional and/or Student Characteristics?*

Two and four-year institutions in Texas were established to serve different populations and have differing sources of governance and resources. Over time, especially with the establishment of Bachelor's programs at 2-year institutions and the catchment area of many regional state institutions being geographically aligned to the 4-year institutions, there is overlap between the students being served. This is especially relevant given the transfer focus of many 2-year institutions that align curriculum and move students through a pipeline to the 4-year institutions. All of the types of basic needs programs were offered at both 2-year and 4-year institutions. There were differences between these two institutional types in terms of the personnel and resource-intensive types of basic needs programs. For example, both on-campus physical and mental health services differed by institutional type, with 2-year institutions being more likely to offer mental health services and 4-year institutions more likely to offer in-person physical health services. Institutions are determining priorities and have demonstrated fiscal response to student needs through the hiring of professional personnel and the allocation of space for these types of services. Likewise, after-school care, while provided at very few institutions, was more likely to be available at 2-year institutions. Like mental and physical health services, childcare services require qualified staff and space.

The associations between student characteristics in terms of the percentage of Pell Grant recipients and the presence of specific types of basic needs programs showed a different pattern. Food pantries were associated with the percentage of Pell Grant recipients after the pandemic but not before. This type of basic needs program is ubiquitous in institutions with students demonstrating greater need but occurred in about two-thirds of institutions with a lower percentage of students receiving Pell Grants. The two types of basic needs programs that were associated with demonstrated student need in terms of Pell Grant receipt were community-referral based. Help to apply for public services was found at almost all institutions, with a higher percentage of students receiving Pell but much rarer, especially prior to the pandemic, at other institutions. The same pattern was seen in physical health referrals to low-cost, off-campus providers.

While much of the research on basic needs programs has focused on student needs, institutional factors, and student body characteristics are relevant in understanding the presence of certain basic needs programs on campus but not others. Given the shared institutional features of being Hispanic-serving institutions in Texas, it may have influenced the establishment of certain basic needs programs. It is unclear if this reflects a normative or mimetic isomorphic institutional response. Future research needs to further examine the growth and spread of basic needs programs and services to understand how they are being implemented and to identify if these types of programs contribute to student success outcomes such as persistence and completion.

*8.3. Texas Hispanic-Serving Institutions and the COVID-19 Pandemic*

This study focused on Texas HSIs. An increasing number of post-secondary institutions are being classified as HSIs, and this represents a rather broad category of institutions with important ingroup differences [16]. In Texas, many HSIs enroll a local student population, and their students reflect socioeconomic conditions within the community. Despite the fact that many of these institutions are often less well-resourced, prior to the pandemic, these institutions had committed to providing services to address unmet student basic needs. Many of the pandemic-era policies had criteria for resource allocation that disadvantaged HSIs [9]. Yet, during the pandemic, the HSIs in Texas continued to provide a variety of basic needs programs for students and anticipated providing those services after the pandemic. This suggests that HSIs are integrating basic needs programs into the array of student services and are connecting these to outcomes important to their institutional mission.

*8.4. Limitations*

There were limitations to this study. First, the presence of certain services on campus does not speak to whether they address the level of need. For example, almost all the institutions had on-campus mental health services, and referrals to mental health services off-campus increased. Yet, other research suggests a lack of access to sufficient mental health services for college students even when some services are provided on campus [54–56]. Second, the study was limited to HSIs in Texas. Future research needs to examine institutional basic needs services across a wider range of institutions, such as institutions in other states. Related to the need to expand this research is the fact that we treat HSIs as a uniform group, yet researchers have identified important differences within this institutional categorization [49,50]. Future research should seek to include a larger sample of HSIs to allow for an examination of differences among HSIs. Finally, in this study, we asked about what the administrators expected to happen after the pandemic. In this late/post-pandemic period, it would be important to examine what basic needs support is being offered on campuses. This relates directly to the need to continue to examine institutional responses to basic needs insecurities among college students.

**9. Conclusions**

While there were changes in the number and type of basic needs programs being offered at the institutions through the pandemic period, the planned resumption of some programs and the addition of others suggests that Texas HSIs are committed to offering these anti-poverty programs on campus. In considering the institutional factors that may promote the establishment or planned provision of specific basic needs programs, the type of institution and a student body with demonstrated financial need was not associated with the presence of most types of basic needs programs. This leaves questions as to what motivates institutions to provide services, especially as many types of basic needs programs are being provided at numerous institutions, and whether these services were important enough to preserve through multiple pivots throughout the pandemic. Future research is needed to continue to examine institutional responses to students' unmet basic needs. Additionally, given limited resources, research needs to evaluate programs and services to determine which improve student success in terms of retention, completion, and other mission-aligned outcomes. It would also be important to establish best practices in postsecondary institutions that serve traditionally underrepresented and economically disadvantaged student populations.

**Author Contributions:** Conceptualization, L.K.Z., T.S. and X.X.; methodology, L.K.Z. and X.X.; software, L.K.Z.; validation, L.K.Z., X.X. and T.S.; formal analysis, L.K.Z., X.X. and S.C.; investigation, L.K.Z.; resources, T.S.; data curation, L.K.Z.; writing—original draft preparation, L.K.Z.; writing—review and editing, L.K.Z., T.S., X.X. and S.C.; visualization, S.C.; supervision, T.S.; project administration, L.K.Z.; funding acquisition, T.S., L.K.Z. and X.X. All authors have read and agreed to the published version of the manuscript.

**Funding:** This work was supported by the ECMC Foundation [Grant #G-1906-12740 | College Success]. The APC was funded by the ECMC Foundation [Grant #G-1906-12740 | College Success].

**Institutional Review Board Statement:** The Institutional Review Boards (IRB) at the University of Tennessee, Knoxville (UTK) and the University of Texas at San Antonio determined that the study did not require IRB oversight and review as defined in 45 CFR 46.102 (3)(1) (UTK IRB-20-06081-XP).

**Informed Consent Statement:** Informed consent was obtained from all participants involved in this project.

**Data Availability Statement:** The data presented in this study are available on request from the corresponding author. The deidentified survey data will be provided without links to the institutional data due to restrictions to ensure participant confidentiality.

**Conflicts of Interest:** The authors declare no conflict of interest.

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
