# Peer review of "Examining Anti-Poverty Programs to Address Student’s Unmet Basic Needs at Texas Hispanic-Serving Institutions over the Course of the COVID-19 Pandemic"

_2813-4346, doi:10.3390/higheredu3010003_

Round 1
Reviewer 1 Report
Comments and Suggestions for Authors
The author(s) have identified an important issue in higher education. Basic needs insecurity has emerged as an important and understudied areas of research. And I am not familiar with studies that explicitly seek to explore basic needs insecurity at HSIs; however, larger studies (such as Crutchfield’s CSU study and work done at the HOPE Center) have included HSIs. I encourage the authors to consider the specific implications related to this context. Overall, I think the manuscript is well-written and the data support the conclusions provided.
Based upon my review of the manuscript, I encourage the author(s) to consider the following recommendations:
· The introduction explains how the COVID context created disruptions for higher education and they seek to understand how that influenced basic needs insecurity. I appreciate why this is important from a research perspective, but I encourage the author(s) to add a couple of sentences explaining why this matters in the post-pandemic context. This would help current readers see how it connects to studies that are being done after the pandemic.
· In the introduction and discussion, encourage the authors to note that some of these trends existed before the pandemic and have been magnified given the economic situation that most postsecondary institutions find themselves with continued reductions in state/federal support as well as the end of the pandemic funding. Well-being and basic needs insecurity (which are related) have been a growing issue for many years and the pandemic magnified those issues, which has had a lingering effect post-pandemic.
· Did the size of the institution matter? Did public/private matter? (I appreciate that the authors may not have these data, but I found myself asking if there was a significant difference based upon these aspects of the institution.)
· The authors ground the study in being focused on Texas HSIs. I wanted a bit more of an exploration of how this context mattered in the discussion. What do these findings mean for HSIs? Is there something the authors could add to the HIS literature? What do these findings mean given the state context? States responded very differently to COVID and the impact on higher education institutions was significantly different when looking at states like California and New York as compared to Texas or Florida.
· This relates a bit to the previous point – I wanted a bit more discussion about what these findings mean and how this trend contributes to our understanding of higher education institutions. I think the data are solid, I just wanted a bit more exploration about how these findings extend current literature.
· Consider reworking the conclusion to provide a more clear summary of the contribution this manuscript makes to the field of higher education. I am not sure we need to focus studies on what motivates institutions to provide these services – that seems to be well understood in the literature. What is uniquely added to the literature by the findings of your study? And what would be the next study that would extend your findings related to basic needs insecurity for students attending HSIs?
Author Response
Please see attached file.
Reviewer #1
Comment 1: The introduction explains how the COVID context created disruptions for higher education and they seek to understand how that influenced basic needs insecurity. I appreciate why this is important from a research perspective, but I encourage the author(s) to add a couple of sentences explaining why this matters in the post-pandemic context. This would help current readers see how it connects to studies that are being done after the pandemic.
- Thank you for this comment. We have revised page 2, para 2 to explain why this matters in a post-pandemic environment. The added statements are lighted in yellow on the draft manuscript and below.
“Addressing student basic needs aligns with institutional student success and completion priorities (Crutchfield & Maguire, 2019; Harris & Wood, 2022; University of Maine System, 2023). The pandemic pivots and subsequent economic and enrollment conditions were disruptive for many broad access HSIs, especially community colleges (Duran & Núñez, 2021). In the post-pandemic, there is growing attention to the costs of attending college, and the financial challenges faced by many students as they pursue post-secondary education. As institutions respond to the basic needs insecurities in a post-pandemic environment, it is important to understand how the pandemic shifted services and programs directed at student’s unmet basic needs. It is unclear if basic needs programs offered prior to the pandemic were determined to be essential services during the pandemic, and therefore necessary to operational and academic continuity. It is critical to have a clear understanding of the changes that occurred to types of basic needs programs provided by institutions over the course of the pandemic because this provides insight into post-pandemic anti-poverty programs that are designed to support student retention and completion.”
Comment 2: In the introduction and discussion, encourage the authors to note that some of these trends existed before the pandemic and have been magnified given the economic situation that most postsecondary institutions find themselves with continued reductions in state/federal support as well as the end of the pandemic funding. Well-being and basic needs insecurity (which are related) have been a growing issue for many years and the pandemic magnified those issues, which has had a lingering effect post-pandemic.
- The authors agree with this comment. A statement has been made in the introduction to clarify the continued natures of these issues. See the highlighted statement on page 1, paragraph 1 and below.
“In the decade prior to the COVID-19 pandemic, there had been growing awareness of the prevalence of basic needs insecurities among students at colleges and universities across the United States. Amid calls to address the economic plight faced by many students within U.S. post-secondary institutions and the results of data gathered by colleges and universities about their students’ food and housing insecurities (Laren et al., 2018), many post-secondary institutions developed new or reconceptualized existing anti-poverty programs to increase the retention and completion among economically disadvantaged students. Often referred to as basic needs initiatives (BNI), these anti-poverty programs sought to reduce non-academic barriers to college persistence and completion (Martinez et al., 2021). The array of services under the umbrella of BNI included food pantries, emergency housing, emergency financial assistance, childcare, transportation, clothing, and healthcare (Larin, 2018; Zottarelli et al., 2022). The COVID-19 pandemic prompted abrupt changes to basic needs programs (Hagedorn-Hatfield et al., 2022). While the trends in awareness and response to student need started before the pandemic, changes made during the pandemic as well as post-pandemic economic conditions have magnified wellness and basic needs insecurities.”
- Additionally, changes have been made to the discussion and are highlighted in yellow in the manuscript.
Comment 3: Did the size of the institution matter? Did public/private matter? (I appreciate that the authors may not have these data, but I found myself asking if there was a significant difference based upon these aspects of the institution.)
- Thank you for this comment. We, too, are interested in further examining institutional factors that may create an organizational response to student needs through the development of specific programs and services, as well as an assessment of those services. We will be continuing this line of research in the future and will include factors such as size (enrollment), urban/suburban/rural, residential/commuter, etc. We will be working to expand the dataset to collect this information in the future.
Comment 4: The authors ground the study in being focused on Texas HSIs. I wanted a bit more of an exploration of how this context mattered in the discussion. What do these findings mean for HSIs? Is there something the authors could add to the HIS literature? What do these findings mean given the state context? States responded very differently to COVID and the impact on higher education institutions was significantly different when looking at states like California and New York as compared to Texas or Florida.
- We have added information about state and system basic needs initiatives on page 4-5 to provide state-level context.
“It is relevant to examine basic needs initiatives to address unmet student basic needs at a state level and aligned to institutional characteristics such as HSI status. The state in which public institutions are located influences institutional policy, priorities, and funding. For examine, California has examined both student need and institutional response to that need across university systems (Crutchfield & Maguire, 2019; Martinez et al., 2021). California has policies, budget allocations, and coordinated support of basic needs programs on college campuses across the state (The California State University, 2023; University of California, 2023). The State University of New York and The City University of New York are engaging in system-wide efforts to address a range of basic needs programs (Young Invincibles, 2023). New Jersey has launched a state-wide basic needs resource page for college students (State of New Jersey, 2023). Texas, in contrast, lacks this kind of organized approach but institutions are encouraged to engage in efforts to support student success, and one approach taken by many Texas colleges and universities has been to develop programs directed at supporting specific student populations, including lower-income and first-generation college students. In the absence of state level initiatives, institutions in states like Texas are still implementing basic needs programming. Understanding the types and prevalence of anti-poverty basic needs programs where it is institutionally driven may provide insights that differ from state-driven initiatives.
In Texas, most community colleges and broad-access regional public institutions are HSIs. Many of the students attending HSIs are from the county or neighboring counties where the institution is located. As such, the HSIs have a significant role in the community and have a student body reflective of that community (Nuñez et al., 2015). Further, HSIs are more likely to enroll first-generation, lower-income, and historically underrepresented, and non/post-traditional students (Cortez, 2018). These students are more likely to experience basic needs insecurities while in college.”
- The authors have also added information in the discussion section to contextualize the importance of HSIs in Texas
“Texas Hispanic-Serving Institutions and the COVID-19 Pandemic
This study focused on Texas HSIs. An increasing number of postsecondary institutions are being classified as HSIs and this represents a rather broad category of institutions with important ingroup differences. In Texas, many HSIs enroll a local student population and their students reflect socioeconomic conditions within the community. Despite the fact that many of these institutions are often less well-resourced, prior to the pandemic, these institutions had committed to providing services to address unmet student basic needs. Many of the pandemic policies to allocate resources to postsecondary institutions during the pandemic, had criteria for resource allocation that disadvantaged HSIs (Miller, 2020). Yet, during the pandemic, the HSIs in Texas continued to provide a variety of basic needs programs for students and anticipated providing those services after the pandemic. This suggests that HSIs are integrating basic needs programs into the array of student services and are connecting these to outcomes important to their institutional mission.”
Comment 5: This relates a bit to the previous point – I wanted a bit more discussion about what these findings mean and how this trend contributes to our understanding of higher education institutions. I think the data are solid, I just wanted a bit more exploration about how these findings extend current literature.
- Thank you for this comment. The authors have made changes to the discussion section and elsewhere in the document. All changes are highlighted in yellow in the manuscript.
Comment 6: Consider reworking the conclusion to provide a more clear summary of the contribution this manuscript makes to the field of higher education. I am not sure we need to focus studies on what motivates institutions to provide these services – that seems to be well understood in the literature. What is uniquely added to the literature by the findings of your study? And what would be the next study that would extend your findings related to basic needs insecurity for students attending HSIs?
- We have added information to the discussion section. It is highlighted in yellow in the manuscript.
- Additionally, we have revised on the conclusion. The revision is highlighted in yellow below and in the manuscript.
“While there were changes in the number and type of basic needs programs being offered at the institutions through the pandemic period, the planned resumption of some programs and the addition of others suggests that Texas HSIs are committed to offering these anti-poverty programs on campus. In considering the institutional factors that may promote the establishment or planned provision of specific basic needs programs, the type of institution and a student body with demonstrated financial need were not associated with the presence of most types of basic needs programs. This leaves questions as to what is motivating institutions to provide services, especially as many types of basic needs programs are being provided at numerous institutions and that these services were important enough to preserve through multiple pivots throughout the pandemic. Future research needs to continue to examine institutional response to students’ unmet basic needs and to evaluate programs and services to determine which improve student success in terms of retention and completion and to establish best practices, especially in postsecondary institutions that serve traditionally underrepresented and economically disadvantaged students.”

Reviewer 2 Report
Comments and Suggestions for Authors
In spite of the fact that the present work presents a very interesting and extremely relevant topic in the academic university environment, such as the provision of basic needs program services during the COVID-19 pandemic, some limitations and parts that could be improved have been detected and are described below.
The abstract does not clarify the conclusions of the work. Only the objective and methodology inherent in the analysis of the provision of 17 types of basic needs programs at Hispanic institutions in Texas throughout the COVID-19 pandemic are stated, with the goal of identifying changes in the number and types of programs offered, as well as factors that may influence the presence of specific types of basic needs programs on campus. It is concluded that the results of this study provide information on institutional provision of basic needs programs by type of institution and demonstrated student need over the course of the COVID-19 pandemic but does not specify any factors that influence the presence of these programs.
Likewise, although the authors in the introduction of the paper specify two research questions "What changes occurred in BNI offerings over the course of the pandemic at Texas Hispanic-serving institutions? Were specific types of basic needs programs associated with institutional and/or student characteristics?" no previous hypotheses are put forward to help understand the reason for their analysis. It is recommended to deepen in the background of the work, to raise a state of the question based on hypotheses, questions and, finally, research objectives.
In the theoretical framework, although a quite specific contextualization of the situation of secondary education in Texas and the needs of Latino students is offered, it is not sufficiently specified what is the socioeconomic situation of the region in the field of tertiary education after the pandemic (it would be interesting to raise some background inherent to data on dropouts for example after the COVID pandemic).
In the methodology the authors clearly explain how the data collection procedure was carried out "Data for this study come from an online survey of college and university administrators at Texas HSIs that was conducted to gather information about the types of basic needs programs offered at their institutions prior to and during the pandemic and the anticipated programs to be offered after the pandemic" as well as the study sample "Four administrators at each Texas HSI were identified from a search of the institution's website". Although the type of questions asked (open-ended and closed-ended) and the form, as well as the response categories (yes, no, not sure) are made clear, there is a lack of pre-questionnaire categories where the authors make it clear what they are analyzing, how many parts the questionnaire is divided into, and what thematic area the questions refer to. This would facilitate the understanding of the instrument and help its evaluation.
In the results, the authors make clear the type of findings reached in terms of outcomes, although the conclusions do not answer the questions posed in the introduction, leaving aside important issues mentioned by the authors: for example, What changes occurred in BNI offerings over the course of the pandemic at Texas Hispanic-serving institutions? Were specific types of basic needs programs associated with institutional and/or student characteristics? It is considered that the second question has not been sufficiently answered, lacking in the work a deep contextualization of the type of student as well as their characteristics in the institutional setting. Likewise, there is no discussion of the factors that influence the presence of these programs, nor is sufficient theoretical contextualization provided.
Author Response
Please see attached file.
Reviewer #2
In spite of the fact that the present work presents a very interesting and extremely relevant topic in the academic university environment, such as the provision of basic needs program services during the COVID-19 pandemic, some limitations and parts that could be improved have been detected and are described below.
- Thank you for the detailed review and the opportunity to revise the manuscript.
Comment 1: The abstract does not clarify the conclusions of the work. Only the objective and methodology inherent in the analysis of the provision of 17 types of basic needs programs at Hispanic institutions in Texas throughout the COVID-19 pandemic are stated, with the goal of identifying changes in the number and types of programs offered, as well as factors that may influence the presence of specific types of basic needs programs on campus. It is concluded that the results of this study provide information on institutional provision of basic needs programs by type of institution and demonstrated student need over the course of the COVID-19 pandemic but does not specify any factors that influence the presence of these programs.
- The authors have revised the abstract. The changes have been highlighted in yellow.
“Many post-secondary institutions have implemented anti-poverty programs to address students’ basic needs insecurities. This study examined the provision of 17 types of basic needs programs at Texas Hispanic-serving institutions over the course of the COVID-19 pandemic with the aim to identify changes in the number and types of programs offered as well as factors that may influence the presence of specific types of basic needs programs on campus. While the average number of basic needs programs per institution varied little over time, the specific types of programs that were offered changed. Institution type as a 2-year or 4-year institution was associated with providing on-campus mental health services, on-campus physical health services, and after school care for students’ children at pre-pandemic and anticipated post-pandemic time points and employing students and free food or meal vouchers at the pre-pandemic time point. The percentage of students receiving Pell Grants was associated with basic needs programs to assist students applying for public services and referrals to off-campus health services pre-pandemic and anticipated post-pandemic. The presence of an on-campus free food pantry was associated with the percentage of students receiving Pell Grants at the anticipated post-pandemic time point only. Texas HSIs offered a range of basic needs programs over the course of the pandemic. Over the course of the pandemic, there were changes to the types of basic needs programs offered. Some types of basic needs programs were associated with institutional and/or student characteristics. Given the continued presence of basic needs programs through the course of the pandemic and into the post-pandemic period, the use of these kinds of programs and services to support students, while influenced by external factors such as the pandemic, appears institutionally established as a way to facilitate going to college for students in need.
Comment 2: Likewise, although the authors in the introduction of the paper specify two research questions "What changes occurred in BNI offerings over the course of the pandemic at Texas Hispanic-serving institutions? Were specific types of basic needs programs associated with institutional and/or student characteristics?" no previous hypotheses are put forward to help understand the reason for their analysis. It is recommended to deepen in the background of the work, to raise a state of the question based on hypotheses, questions and, finally, research objectives.
- The authors agree with Reviewer 2’s assessment. We have added the following to the manuscript. This is available starting at the bottom of page 4 and is highlighted in yellow.
“It is relevant to examine basic needs initiatives to address unmet student basic needs at a state level and aligned to institutional characteristics such as HSI status. The state in which public institutions are located influences institutional policy, priorities, and funding. For examine, California has examined both student need and institutional response to that need across university systems (Crutchfield & Maguire, 2019; Martinez et al., 2021). California has policies, budget allocations, and coordinated support of basic needs programs on college campuses across the state (The California State University, 2023; University of California, 2023). The State University of New York and The City University of New York are engaging in system-wide efforts to address a range of basic needs programs (Young Invincibles, 2023). New Jersey has launched a state-wide basic needs resource page for college students (State of New Jersey, 2023). Texas, in contrast, lacks this kind of organized approach but institutions are encouraged to engage in efforts to support student success, and one approach taken by many Texas colleges and universities has been to develop programs directed at supporting specific student populations, including lower-income and first-generation college students. In the absence of state level initiatives, institutions in states like Texas are still implementing basic needs programming. Understanding the types and prevalence of anti-poverty basic needs programs where it is institutionally driven may provide insights that differ from state-driven initiatives.
In Texas, most community colleges and broad-access regional public institutions are HSIs. Many of the students attending HSIs are from the county or neighboring counties where the institution is located. As such, the HSIs have a significant role in the community and have a student body reflective of that community (Nuñez et al., 2015). Further, HSIs are more likely to enroll first-generation, lower-income, and historically underrepresented, and non/post-traditional students (Cortez, 2018). These students are more likely to experience basic needs insecurities while in college.
Given Texas’ lack of a state-wide or postsecondary system-wide basic needs initiatives for college students, the economic situation of many Texas post-secondary students, and the importance of HSIs in providing post-secondary educational opportunities to local students, we first examined the prevalence of basic needs programs at Texas HSIs as we answered the question: What changes occurred in BNI offerings over the course of the pandemic at Texas Hispanic-serving institutions? To do so, we explored changes in the number and types of basic needs programs offered before the pandemic and during the pandemic. Then we examined changes to the number and types of basic needs programs that institutions planned to support after the pandemic.
Next, we examined two factors that may influence the presence of specific basic needs programs at post-secondary institutions. We focused on one institutional factor, classification of the institution as a 2-year or 4-year institution. This is relevant given that student body characteristics differ across these two institution types and the funding sources for the two types of institutions in Texas differ. We used percentage of students receiving Pell grants as the student-focused factor given that it indicates the economic conditions of the student body. We asked the question: Were specific types of basic needs programs associated with institutional and/or student characteristics? This question was answered through the testing of two hypotheses for each type of basic needs programs identified in the study.
Hypotheses 1: There is a relationship between institution type (e.g., 2-year and 4-year and the presence of [specific basic needs program].
Hypothesis 2: There is a relationship between the percentage of students receiving Pell grants and the presence of [specific basic needs program].”
Comment 3: In the theoretical framework, although a quite specific contextualization of the situation of secondary education in Texas and the needs of Latino students is offered, it is not sufficiently specified what is the socioeconomic situation of the region in the field of tertiary education after the pandemic (it would be interesting to raise some background inherent to data on dropouts for example after the COVID pandemic).
- The authors have added a paragraph on page 3 describe enrollment and economic conditions within Texas colleges and universities from prior to the pandemic to the most recent year available.
“Enrollment at Texas public institutions has declined over the course of the pandemic. In academic year 2018-2019, there were 1,570,539 students enrolled in Texas nonprofit higher education institutions (Texas Higher Education Coordinating Board, 2020). In the first full academic year of the pandemic, 2020-2021, enrollment declined to 1,509,761 (Texas Higher Education Coordinating Board, 2022a), and then continued to decline in 2021-2022 to 1,490,079 students (Texas Higher Education Coordinating Board, 2023. Since before the pandemic, the Texas Higher Education Coordinating Board (2020, 2022a, 2023) finds that students attending Texas post-secondary institutions rely extensively on federal financial assistance and experience significant unmet economic need every year. Over this time period, approximately 20% unduplicated financial aid recipients were from households with income below the poverty line. Over a third of unduplicated financial aid recipients were from households under the median income. The state had the second highest food insecurity rate in the United States, which had increased in 2022 to almost 1 in 6 households being food insecure (Rabbitt et al., 2023). There was evidence of unmet student basic needs on campus prior to and during the pandemic (Abu & Oldewage-Theron, 2019; Biediger-Friedman et al., 2016; Brito-Silva et al., 2022; Henry et al., 2023; Manboard et al., 2016). Given the economic conditions of many students’ households throughout the pandemic and into the post-pandemic, it is reasonable to expect that unmet needs will continue to need to be addressed.”
Comment 4: In the methodology the authors clearly explain how the data collection procedure was carried out "Data for this study come from an online survey of college and university administrators at Texas HSIs that was conducted to gather information about the types of basic needs programs offered at their institutions prior to and during the pandemic and the anticipated programs to be offered after the pandemic" as well as the study sample "Four administrators at each Texas HSI were identified from a search of the institution's website". Although the type of questions asked (open-ended and closed-ended) and the form, as well as the response categories (yes, no, not sure) are made clear, there is a lack of pre-questionnaire categories where the authors make it clear what they are analyzing, how many parts the questionnaire is divided into, and what thematic area the questions refer to. This would facilitate the understanding of the instrument and help its evaluation.
- The authors have added the requested details to the text on page 6, paragraph 2. The additions are highlighted below and in the manuscript.
“The survey consisted of a series of close- and open-ended questions. The instrument was divided into two parts. The first part focused on specific basic needs initiatives and included nine thematic areas: food, transportation, housing, financial assistance, clothing, childcare, mental and physical health services, and social service referrals, and employment. The themes identified were found in the academic and practice literature on basic needs programs. Respondents were asked about a total of 17 basic needs programs within the nine themes. For each basic needs item, respondents were asked a set of three questions. First, “Prior to the pandemic, did your institution have [basic needs program] for students in need?” Response categories were “yes”, “no”, and “not sure”. This was followed by the question, “This semester, do you currently have [basic needs program] for students in need?” Response categories were “yes”, “no”, and “not sure”. Then, participants were asked, “In the future, do you anticipate providing [basic needs program] to students in need?” The response categories were “definitely yes”, “probably yes”, “probably not” and “definitely not”.
In the second part of the survey, participants were also asked to provide their institutional affiliation….”
Comment 5: In the results, the authors make clear the type of findings reached in terms of outcomes, although the conclusions do not answer the questions posed in the introduction, leaving aside important issues mentioned by the authors: for example, What changes occurred in BNI offerings over the course of the pandemic at Texas Hispanic-serving institutions? Were specific types of basic needs programs associated with institutional and/or student characteristics? It is considered that the second question has not been sufficiently answered, lacking in the work a deep contextualization of the type of student as well as their characteristics in the institutional setting. Likewise, there is no discussion of the factors that influence the presence of these programs, nor is sufficient theoretical contextualization provided.
- The authors agree and have corrected the omission. The addition text starts on page 14, paragraph 2. It is also provided below.
“Were Specific Types of Basic Needs Programs Associated with Institutional and/or Student Characteristics?
Two and four year institutions in Texas were established to serve different populations, and have differing sources of governance and resources. Over time, especially with the establishment of bachelors programs at 2-year institutions and the catchment area of many regional state institutions being geographically aligned to the 4-year institutions, there is overlap between the students being served. This is especially relevant given the transfer focus of many 2-year institutions that align curriculum and move students through a pipeline to the 4-year institutions. All of the types of basic needs programs were offered at both 2-year and 4-year institutions. There were differences between these two institutional types in terms of the personnel and resource intenstive types of basic needs programs. For example, both on-campus physical and mental health services differed by institutional type, with 2 year institutions being more likely to offer mental health services and 4-year institutions more likely to offer in-person physical health services. Institutions are determining priorities and have demonstrated fiscal response to student needs through the hiring of professional personnel and allocation of space for these types of services. Likewise, after school care, while provided at very few institutions, was more likely to be available at 2-year institutions. Like mental and physical health services, childcare services require qualified staff and space.
The associations between student characteristics in terms of percentage of Pell grant recipients and the presence of specific types of basic needs programs showed a different pattern. Food pantries were associated with percentage of Pell grant recipients after the pandemic but not before. This type of basic needs program is ubiquitous in institutions with students demonstrating greater need but occurred in about two-thirds of institutions with a lower percentage of students receiving Pell grants. The two types of basic needs programs that were associated with demonstrated student need in terms of Pell grant reciept were community-referral based. Help applying for public services was found at almost all institutions with a higher percentage of students receiving Pell but much rarer, especially prior to the pandemic, at other institutions. The same pattern was seen in physical health referrals to low cost off-campus providers.
While much of the research on basic needs programs has focused on student need, institutional factors and student body characteristics are relevant in understanding the presence of certain basic needs programs on campus but not others. Given the shared institutional features of being Hispanic-serving institutions in Texas may have influenced the establishment of certain basic needs programs. It is unclear if this reflects a normative or mimetic isomorphic institutional response. Future research needs to further examine the growth and spread of basic needs programs and services to understand how they are being implemented and to identify if these types of programs contribute to student success outcomes such as persistence and completion.
Texas Hispanic-Serving Institutions and the COVID-19 Pandemic
This study focused on Texas HSIs. An increasing number of postsecondary institutions are being classified as HSIs and this represents a rather broad category of institutions with important ingroup differences (Nuñez et al., 2015). In Texas, many HSIs enroll a local student population and their students reflect socioeconomic conditions within the community. Despite the fact that many of these institutions are often less well-resourced, prior to the pandemic, these institutions had committed to providing services to address unmet student basic needs. Many of the pandemic-era policies had criteria for resource allocation that disadvantaged HSIs (Miller, 2020). Yet, during the pandemic, the HSIs in Texas continued to provide a variety of basic needs programs for students and anticipated providing those services after the pandemic. This suggests that HSIs are integrating basic needs programs into the array of student services and are connecting these to outcomes important to their institutional mission.”

Round 2
Reviewer 2 Report
Comments and Suggestions for Authors
Based on the changes included by the authors in the indicated sections (abstract, theoretical framework, methodology, results and conclusions), I consider that sufficient improvements have been made -removed for peer-review-. I believe that the authors, with these changes, have significantly improved the content and structure of the paper, providing a much clearer and more concrete analysis based on their article, which is interesting and of great value to the academic community.